# Association between Anti-Müllerian Hormone Concentration and Inflammation Markers in Serum during the Peripartum Period in Dairy Cows

**DOI:** 10.3390/ani11051241

**Published:** 2021-04-26

**Authors:** Hiroaki Okawa, Danielle Monniaux, Chihiro Mizokami, Atsushi Fujikura, Toshihiro Takano, Satoko Sato, Urara Shinya, Chiho Kawashima, Osamu Yamato, Yasuo Fushimi, Peter L. A. M. Vos, Masayasu Taniguchi, Mitsuhiro Takagi

**Affiliations:** 1United Graduate School of Veterinary Medicine, Yamaguchi University, Yamaguchi 753-8515, Japan; k2414076@kadai.jp (H.O.); masa0810@yamaguchi-u.ac.jp (M.T.); 2Fukuoka Prefecture Dairy Cooperative Association, Fukuoka 839-0832, Japan; mizokamic@gmail.com (C.M.); fuku-sinryou@f-rakukyou.or.jp (A.F.); t-takano@f-rakukyou.or.jp (T.T.); 3Guardian Co. Ltd., Kagoshima 890-0033, Japan; yasuo243@gmail.com; 4Physiologie de la Reproduction, Centre INRA, 37380 Nouzilly, France; danielle.monniaux@gmail.com; 5Soo Agriculture Mutual Aid Association, Soo 899-8212, Japan; satoko@nosai-soo.com (S.S.); urara@nosai-soo.com (U.S.); 6Field Center of Animal Science, Obihiro University of Agriculture and Veterinary Medicine, Obihiro 080-8555, Japan; kawasima@obihiro.ac.jp; 7Joint Faculty of Veterinary Medicine, Kagoshima University, Kagoshima 890-0065, Japan; osam@vet.kagoshima-u.ac.jp; 8Department Population Health Sciences, Farm Animal Health, Section Reproduction, Utrecht University, Yalelaan 7, 3584 CL Utrecht, The Netherlands; P.L.A.M.Vos@uu.nl

**Keywords:** anti-Müllerian hormone, dairy cattle, endometritis, inflammation, reproductive efficacy

## Abstract

**Simple Summary:**

Changes in anti-Müllerian hormone (AMH) levels, a fertility marker in dairy cows, during the peripartum period associated with reproductive recovery have not been fully evaluated. We investigated the relationship between changes in AMH concentration and inflammation markers in serum during the peripartum period in dairy cows. We found a relationship between changes in AMH concentration, especially reflected in the AMH ratio during the perinatal period, and the systemic inflammation status of dairy cows. Excessive inflammation during the early postpartum period may decrease AMH levels and subsequently affect the reproductive prognosis of postpartum cows. Elucidating the mechanism of perinatal AMH changes and the beneficial effects of AMH may improve reproductive efficacy in the dairy industry.

**Abstract:**

The relationships between changes in anti-Müllerian hormone (AMH) concentration and various traits, including milk somatic cell counts (SCC), were evaluated. Blood samples were collected from 43 Holstein cows 14 days before (D-14) and 10 (D10) and 28 days after (D28) parturition, and vaginal discharge score (VDS) and polymorphonuclear leukocyte (PMNL) percentages were assessed in endometrial samples at D28. Cows were separated into four quartiles (Q1–Q4) based on changes in AMH concentration during the peripartum period (AMH ratio: D28/D-14). Correlations between AMH ratio and each parameter were evaluated and classified into high-AMH (Q4, 1.83 ± 0.12, *n* = 11) and low-AMH (Q1, 0.83 ± 0.05, *n* = 11) groups. The AMH ratio was positively correlated with magnesium and non-esterified fatty acids levels, and the albumin/globulin ratio at D10 and D28, but negatively correlated with serum amyloid A (SAA) at D10. SAA and γ-globulin levels were significantly higher in the low-AMH group at D28. There was no significant difference in VDS, PMNL percentage, and milk SCC between the two groups. The decreasing AMH ratio from the prepartum to the postpartum period corresponds to high inflammation biomarker levels. Whether it subsequently affects the reproductive prognosis of postpartum cows needs further investigations.

## 1. Introduction

Optimal reproductive efficiency significantly influences the dairy industry, and it is an essential factor for the management of dairy farms. Therefore, reliable biomarkers with high variability, repeatability, heritability, and association with reproductive performance of cattle are essential [1,2]. Field studies have investigated various traits, including dietary managements, rearing environments, genetic selection [3], and disease status of both the systemic and genital tract, especially during the perinatal period, the most critical period for preparation and establishment of next pregnancy in dairy cows. In this regard, studies have aimed to elucidate the relationship between metabolic status (such as negative energy balance (NEB)) and the uterus (including parturition such as dystocia and retained placenta) during the perinatal period, and that between postpartum ovarian function and/or reproductive efficacy of cows [3,4]. We recently reported the efficacy of vaginal discharge score (VDS) in diagnosis of clinical endometritis and demonstrated that the VDS grading system improved dairy herd reproductive performance [5]. Additionally, the occurrence of VDS Grade = 1 (evaluated as mucus containing flecks of white or off-white purulent material) or calving abnormality (dystocia, stillbirth, twining, and retained placenta) might affect reproductive performance [6]. 

Dairy cows in perinatal periods have been reported to face NEB through production of non-esterified fatty acids (NEFAs) and oxidative stress (as indicated by the increased production of reactive oxygen species (ROS)), together with digestive acidosis and social stress, which may act as proinflammatory factors [7]. Recent studies have focused on the relationship between postpartum ovarian follicular functions and uterine and systemic inflammation status during perinatal period through evaluation of the number of polymorphonuclear leukocyte (PMNL) cells via endometrial cytological examinations, performance of metabolic profile test (MPT), and determination of the concentration of acute phase proteins (APPs) such as albumin (Alb), haptoglobin (HG), and paraoxonase (PON) [3,8]. Moreover, uterine bacterial infection (inflammation) has been reported to perturb ovarian follicular growth and function, leading to accumulation of bacterial lipopolysaccharide (LPS) in the follicular fluid, which in turn leads to an increase in follicular atresia and granulosa cell apoptosis [7,9]. This may not only affect preantral and antral follicular functions, but also antral follicle count (ovarian reserve) in the postpartum period of cows [7,10]. 

Anti-Müllerian hormone (AMH) is a glycoprotein belonging to the transforming growth factor-beta superfamily and secreted by ovarian granulosa cells primarily from pre-antral and early antral follicles of females [11,12]. Recent studies have shown that the blood AMH level in cattle varies in individual cows, and that it can be a reliable endocrine marker for the number of ovulation events and embryos produced in response to superovulation or via ovum pick-up and in vitro production [13,14,15,16]. AMH is an endocrine marker closely associated with the gonadotrophin-responsive ovarian reserves, and with the size of the pool of growing preantral and small antral follicles [12]. Monniaux et al. [12] demonstrated that, although high, long-term, and intraindividually consistent plasma AMH levels are reported in cattle, some variations in AMH concentration exist throughout the reproductive life of animals, especially from the gestation to the postpartum period. In vivo changes in AMH ovarian production are reflected by AMH endocrine changes. The growth of small antral follicles, which produce AMH, can be influenced by changes in metabolic hormone concentrations induced by acute changes in nutrient intake [17,18,19]. These follicles are also capable of responding to inflammatory mediators, which may perturb their development [20,21]. However, direct evidence of immunological or other health conditions detrimental to reproductive performance and influencing the number or dynamics of AMH-producing follicles has not been demonstrated so far. Moreover, to our knowledge, no reports are available on the relationship between the changes in serum AMH concentration and inflammation status derived from both systemic NEB and bacterial infections of the genital tract that occur during the perinatal period in cows. We hypothesized that changes in inflammation status caused by the above-mentioned factors during perinatal periods in cows will affect the serum AMH concentration and are associated with the postpartum reproductive performance or efficacy of the cows. 

The objective of this study was to evaluate the relationship between changes in serum AMH concentration and various traits, including metabolic and inflammatory blood profiles, recovery of reproductive organs (monitoring of both VDS and PMNL), and subsequent reproductive performance of cows during the peripartum period. Additionally, the level of serum amyloid A (SAA), which is one of the most reliable APPs primarily produced by the liver and other tissues, including the uterus [22,23], was determined to describe inflammation status and its relationship with AMH variations.

## 2. Materials and Methods

The experiments were conducted according to the regulations concerning the protection of experimental animals and the guidelines of Yamaguchi University, Japan (No. 40, 1995; approval date: 27 March 2017), and informed consent was obtained from the farmers.

### 2.1. Animals and Management

The study was conducted using 43 randomly selected Holstein Friesian cows after clinically normal calving from November 2018 to July 2019, derived from four commercial dairy herds (approximately 10 cows per herd) in Fukuoka Prefecture, Japan, where ambient mean temperature during the sampling period ranged between 5.7 °C at minimum in January and 26.9 °C at maximum in July. The size of four lactating herds varied from 35 to 50 cows. The herds were non-seasonal and were milked twice daily, and average milk production varied between 9500 and 10,500 kg/cow/year. Throughout the experimental period, the cows from the four dairy herds were fed a diet mainly consisting of grass, whole crop silage as roughage, and concentrate for dry cows (dry matter basis: 102–137 g of crude protein/kg and 5.4–6.0 MJ of NEL/kg) or dairy cows (dry matter basis: 145–159 g of crude protein/kg and 6.9–7.4 MJ of NEL/kg). The cows were fed ad libitum in accordance with the Japanese feeding standard of dairy cattle (Agriculture, Forestry and Fisheries Research Council Secretariat, 2017) to meet their maintenance, growth, and lactation requirements. One herd was fed with total mixed ration, while three herds were fed with a separate feed mixture based on instructions of a nutritionist from Fukuoka Prefecture Dairy Cooperative Association. The parity of the cows ranged from one to seven, with no significant differences between the four herds (mean ± SEM; 3.0 ± 0.2; *n* = 43). All cows were managed in a tie stall with rubber mattresses, and the estrous detection was based on hyperemia and swelling of vulva, mucus discharge, bellowing, and restlessness. Cows were inseminated 8 to 14 h after appearance of these signs by local artificial insemination (AI) technicians. The voluntary waiting period in each herd was set at 60 days postpartum. Monthly follow-ups were conducted for reproductive examination, including treatment of reproductive disorders and pregnancy diagnosis. Pregnancy was confirmed through transrectal palpation or ultrasonography 40 days after the previous insemination. For cows not inseminated within 80 days postpartum or not pregnant, additional treatment was performed, such as a timed AI program using combined injection of PGF_2α_ and estradiol. Additionally, all postpartum metabolic and/or reproductive diseases were diagnosed by a managing veterinarian based on clinical signs or symptoms, such as ketosis, milk fever, indigestion, and mastitis, before blood sampling was conducted at 28 days after calving was recorded. The summary of the experimental protocol is shown in Figure 1.

### 2.2. Blood Sample Collection and Analysis

Samples were obtained three times in 14 days prior to expected parturition (D-14) and 10 (D10) and 28 days (D28) after parturition of each cow. Blood samples were collected by caudal venipuncture at 2–3 h after morning feeding. Additionally, the body condition score (BCS) was evaluated on a 5-point scale with increments of 0.25, as described by Ferguson et al. [24], concomitant with the rumen fill score (RFS), recorded as described by Kawashima et al. [25] at D-14 and D28. The samples were immediately placed in a box on ice for cooling and protection from light and were transported to a laboratory. After centrifugation at 500× *g* for 15 min at room temperature, the serum samples were frozen at −30 °C until further analysis.

For assessment of the metabolic profiles, serum biochemical analysis was performed (measured using Labospect 7180 autoanalyzer; Hitachi, Tokyo, Japan) to determine the following measures, according to a previous report [26]: blood glucose (Glu), NEFA, total cholesterol (T-Cho), triglyceride (TG), 3-hydroxybutyric acid (3-HB), total protein (TP), albumin (Alb), albumin/globulin ratio (A/G), bilirubin (Bil), urea nitrogen (BUN), aspartate aminotransferase (AST), γ-glutamyltransferase (GGT), calcium (Ca), magnesium (Mg), iron (Fe), and inorganic phosphorus (iP). SAA concentration was also measured using an automated biochemical analyzer (Pentra C200; HORIBA ABX SAS, Montpellier, France) with a special SAA reagent for animal serum or plasma (VET-SAA ‘Eiken’ reagent; Eiken Chemical Co. Ltd., Tokyo, Japan). SAA concentration was calculated using a standard curve generated using a calibrator (VET-SAA calibrator set; Eiken Chemical Co. Ltd., Tokyo, Japan). Additionally, the serum IGF1 concentration was determined by enzyme immunoassay using the biotin-streptavidin amplification technique for evaluation of hepatic metabolism, according to a previous report [27].

AMH concentration was measured using a bovine AMH ELISA kit (AnshLabs, Webster, TX, USA), according to a previous report [16]. Undiluted plasma (50 μL) was used for the assay, and the assay had a limited detection of 11 pg/mL and a coefficient of variation of 2.9% according to the manufacturer’s instructions. In the present study, the paired serum samples collected at D-14 and D28 from the same cow were analyzed with the same assay to prevent inter-assay difference. 

The concentration of serum protein fractions was estimated from their ratio using cellulose acetate electrophoresis and the level of total serum protein. Cellulose acetate electrophoresis was conducted using an automatic electrophoresis device (CTE9800, Joko, Tokyo, Japan) with SELECA-VSP membrane (Advantech Co., Ltd., Tokyo, Japan). 

### 2.3. Evaluation of Ovary and Uterine Condition

Prior to sampling at D28, ultrasound scanning of both ovaries was performed to confirm the presence of corpus luteum as previously reported [28], and VDS was determined using a Metricheck device (Simcro Tech Ltd., Hamilton, New Zealand) that comprised a stainless-steel rod and a silicon cup. The VDS assessment method was similar to the method described, with slight modifications as follows: VDS0 = clear or translucent mucus; VDS1 = mucus containing flecks of white pus or off-white purulent material; VDS2 = discharge containing < 50% purulent material; and VDS3 = discharge containing > 50% purulent material [5]. Concomitant with these samples, endometrial samples were collected using a cytobrush (Metribrush; Fujihira Industry Co., Ltd., Tokyo, Japan) as previously described [29]. The cytobrush was attached onto a 611 mm-long plastic rod and placed in a solid stainless-steel tube (550 mm in length and 4 mm in diameter) for passage through the cervix at D28. The vulva was washed, wiped using a paper towel, and the instrument was inserted through the vagina and advanced through the cervix into the body of the uterus. The brush was then ejected from the steel tube, and endometrial samples were collected by rotating the brush while in contact with the endometrium. The brush was then retracted into the steel tube prior to removal from the uterus. Cytology slides were prepared by rolling the brush containing the tissue samples onto two clean microscopic slides. The slides were then fixed, stained with Diff Quik Solution (Sysmex Corp., Kobe, Japan), washed, and air-dried. Cytological assessment was conducted to determine the PMNL percentage by counting a minimum of 200 nucleated cells, as previously described [29].

### 2.4. Somatic Cell Count of Milk 

Somatic cell count (SCC) of each examined cow was obtained from farm records and was routinely collected for the dairy herd in the progeny test at the beginning of each month. Milk samples were collected in 30 mL plastic tubes with preservative using specific sampling devices during milking. The samples were sent to the laboratory of Fukuoka Prefecture Dairy Cooperative Association, and SCC was determined with a fluoro-optical method using Fossomatic FC (FOSS Ltd., Tokyo, Japan). First and second sets of postpartum SCC data collected within 60 days after parturition and assessed to monitor the mastitis status in each cow. In the present study, cows scoring SCC higher than 200,000 cells/mL were considered to have subclinical mastitis, based on previous reports [30,31]. Additionally, the present study also assessed the record for the treatment of mastitis from each examined dairy farm.

### 2.5. Reproductive Records

Reproductive performance data were compared between groups and included the following: calving to first AI (FAI) interval, successful conception after FAI, the number of AI required to become pregnant, the number of days open, and the number of pregnant cows within 200 days postpartum. Reproductive performance data were collected until confirmation of pregnancy or culling. 

### 2.6. Data Management and Classification of the Cows Based on Their AMH Concentration

For the first evaluation, correlations between the change in AMH concentration during the peripartum period from D-14 to D28 (AMH ratio: D28/D-14) and the results of each serum metabolic and inflammation parameter at D-14, D10, and D28, as well as both VDS and PMNL at D28, were evaluated in 43 examined cows to identify influencing factors. For the second evaluation, cows were classified into quartiles (Q1, Q2, Q3, and Q4) based on the AMH ratio. Q4, Q3, Q2, and Q1 were selected as the high-AMH (H-AMH) (*n* = 11), high-medium-AMH (HM-AMH) (*n* = 11), low-medium-AMH (LM-AMH) (*n* = 10), and low-AMH (L-AMH) (*n* = 11) groups, respectively, as shown in Figure 1.

### 2.7. Statistical Analysis

Data were expressed as the mean ± SEM or as a percentage. Statistical analyses were performed using BellCurve for Excel software program (Social Survey Research Information Co., Ltd., Tokyo, Japan). Association of AMH ratio of all 43 cows with each serum metabolic and inflammation parameter at D-14, D10, and D28, as well as with both VDS and PMNL% at D28, was evaluated using Pearson’s correlation coefficient. Additionally, the results of body condition score (BCS), rumen fill score (RFS), and serum biochemical variables were compared between the H-AMH and L-AMH groups using Student’s *t*-test. The results of the outcomes, serum protein fractions, SAA, serum AMH concentration, milk SCC, reproductive variables, and the number of medications were compared between the H-AMH and L-AMH groups using a linear mixed model with the farm as the random factor and the lactation number as the covariate. Pregnancy, culling rates, and the number of cows that required medications were compared using a generalized linear mixed model (link function: logit). Since pre-power analysis was not performed in this study, the power was calculated from the actually obtained sample size by post hoc power analysis when the effect size was medium or large for comparison test. A *p*-value < 0.05 was considered statistically significant, whereas *p*-values ranging between 0.05 to 0.1 were considered to indicate a trend towards significance.

## 3. Results

### 3.1. First Evaluation: Relationship between the AMH Ratio (AMH at D28/AMH at D-14) with Serum Metabolic and Inflammation Parameters

The main correlation values found between the AMH ratio with serum metabolic and inflammation parameters at D-14, D10, and D28 in 43 cows are shown in Table 1. No significant correlation was observed between any of the examined parameters at D-14, except for TP. Significant positive correlations were observed between the AMH ratio and A/G ratio, Mg, and NEFA; however, a negative correlation was observed between the AMH ratio and SAA at D10. Additionally, significant positive correlations were observed between the AMH ratio and A/G ratio, Mg, and NEFA, and a negative correlation was observed between the AMH ratio and TP at D28. No correlation was observed between the AMH ratio and other examined parameters at D10 and D28. Representative parameters that showed significant correlation with the AMH ratio are shown in Figure 2. 

### 3.2. Second Evaluation: AMH Ratio and Serum Biochemical Markers

Serum AMH concentrations at D-14 and D28 and the AMH ratio (D28/D-14) of each examined cow (*n* = 43) are shown in Figure 3. Table 2 shows the results of the mean AMH concentration at D-14 and D28 and the mean AMH ratio of each group of cows. In this study, the D-14 sample was obtained 2 weeks before the expected delivery date; therefore, the date of sampling before delivery and the number of days until the actual delivery date were assessed. There was no significant difference in the number of days from D-14 sampling to calving among Q1 (16.6 days), Q2 (15.9 days), Q3 (10.8 days), and Q4 (13.7 days). No significant difference was observed in the AMH concentration among Q1, Q2, Q3, and Q4 groups at D-14 and D28. Additionally, no significant difference was observed in the parity among Q1 (3.3 ± 0.4), Q2 (2.9 ± 0.3), Q3 (2.7 ± 0.4), and Q4 (3.0 ± 0.6). As expected, when the AMH ratios were compared, a significant difference (*p* < 0.05) was observed between the H-AMH (Q4, *n* = 11, 1.83 ± 0.12) and L-AMH (Q1, *n* = 11, 0.83 ± 0.05) groups. Based on the results from the first evaluation, we focused on the cows classified into the H- and L-AMH groups to determine factors that may affect AMH concentrations. 

Table 3 shows the results of body condition score (BCS), rumen fill score (RFS), and serum biochemical analyses (mean ± SEM) of H- and L-AMH cows at D-14 and D28. At D-14, BCS of the H-AMH group tended to be lower (*p* < 0.1) than that of the L-AMH group, but no significant difference was observed at D28. At both D-14 and D28, TP was significantly lower (*p* < 0.05) in the H-AMH group than in the L-AMH group. At D28, concentrations of Mg, 3-HB, Alb, and the A/G ratio were significantly higher in the H-AMH group than that in the L-AMH group (*p* < 0.05; as shown in Table 3). (Appendix A).

As significant differences in TP and A/G ratio were observed in the present study, concentrations of serum protein fractions were determined and are shown together with the results of SAA concentrations in Table 4. At D28, Alb concentrations in the H-AMH group were significantly higher (*p* = 0.007) than those in the L-AMH group, but SAA concentrations in the L-AMH group were significantly higher (*p* = 0.022) than those in the H-AMH group. α_1_-globulin concentrations in the L-AMH group at D-14 were significantly higher (*p* = 0.032) than those in the H-AMH group. Additionally, β- and γ-globulin concentrations in the L-AMH group at D28 were significantly higher (*p* = 0.027 and 0.044, respectively) than those in the H-AMH group. Figure 4 shows the results of protein electrophoresis, which include representative samples from both the H-AMH (AMH ratio: 2.9) and L-AMH (AMH ratio: 0.5) groups. Samples from the L-AMH group showed typical higher values of β- and γ-globulin fractions compared with those in the samples from the H-AMH group.

### 3.3. Second Evaluation: AMH Ratio, Clinical Status of Mammary Gland and Reproductive Organs, and Reproductive Records

Table 5 shows the results of SCC in milk and postpartum reproductive traits of recovery of uterine and ovarian function. No significant difference was observed in SCC, PMNL, VDS, and CL rate between the H-AMH and L-AMH groups in the present study. 

Table 6 shows the results of the number of postpartum clinical treatments as well as postpartum reproductive records of the H-AMH and L-AMH groups. The number of cows that required treatment by D28 postpartum was significantly lower in the H-AMH group than in the L-AMH group (3/11 vs. 9/11; *p* = 0.021). In addition, the average number of medications tended to be lower in the H-AMH group than in the L-AMH group (0.8 (0.0 to 3.2) vs. 3.9 (1.5 to 6.3); *p* = 0.072).

No significant difference was observed in calving to first AI interval, pregnancy rate of first AI, number of days open, and number of AI to pregnancy. However, the number of pregnant cows within 200 days postpartum in the H-AMH group was higher (*p* = 0.039) than that in the L-AMH group. Additionally, the culling rate in the herds of H-AMH cows was lower (*p* = 0.010) than that of L-AMH cows. 

According to the post hoc power analysis, the detection power in comparison between H-AMH (*n* = 11) and L-AMH (*n* = 11) was 0.607 when the effect size was large (effect size: d = 1.0).

## 4. Discussion

The smooth recovery/resumption of uterine and ovarian activities plays a crucial role in fertility after calving in dairy cows. Previous reports have demonstrated the effects of various diseases on reproduction in dairy herds [32,33]. Ribeiro et al. [33] reported that disease onset at the preantral or antral stage of ovulatory follicular development had detrimental effects on oocyte competence and pregnancy results and impaired the uterine environment. Thus, the aim of the present study was to evaluate the importance of monitoring postpartum ovarian reserves by studying the relationship between change in serum AMH and various traits, which brings an insight into the metabolic status; systemic, uterine, and mammalian inflammation status; and subsequent clinical reproductive performance of dairy cows. We demonstrated that inflammation markers during early postpartum were associated with a decrease in AMH levels, suggesting that inflammation affected the dynamics of the AMH-secreting follicles.

AMH measurement in agricultural species, especially for cattle production, has been proven to be a reliable, cost-effective, easy-to-assess practical marker for fertility, and would be highly desirable [15,34,35]. It has been reported that beneficial clinical application possibilities include multiple ovulations, embryo transfer (MOET), and ovum pick-up and in vitro production (OPU-IVP) protocols [16,36,37]. However, as demonstrated by Monniaux et al. [12], some variations in AMH concentration exist during the gestation and postpartum periods in cattle, with possible regulatory factors such as gonadotrophins, GH, insulin, IGF1, or NEFA reflecting the metabolic status during perinatal period. Although a significant positive correlation was observed between the AMH ratio and both NEFA and Mg in the first evaluation, in the present study, no difference was observed in IGF1 levels between the H-AMH and L-AMH groups. A significant difference was observed in the concentrations of Mg and 3-HB between the two groups. Acetone, acetoacetate, and 3-HB are considered ketones produced by metabolism of NEFAs and volatile fatty acids [38]. In the present study, although BCS tended to be different (H-AMH group: 3.2 ± 0.1, L-AMH group: 3.4 ± 0.1, *p* < 0.1) during the dry period (D-14), no differences were observed in both BCS and RFS, concomitant with NEFA and glucose concentrations during postpartum (D28), and none of the groups was under the NEB. 3-HB is mainly derived from butyrate metabolism in the digestive organs and is well known as an indicator of feed intake and rumen development in cattle [39]. Cows may have simultaneously high 3-HB and low NEFA concentrations that originate from their diet [40]. Although a previous report suggested that the relationship between lower Mg concentration and occurrence of endometritis might be due to hepatocellular damage during early lactation [41], we did not observe any abnormal values for the traits of hepatic damage and function in the present study. Therefore, the higher 3-HB concentration in the H-AMH group with significant correlation with NEFA in the first evaluation observed in the present study may reflect general health condition. High intake of dry matter promotes the growth of rumen microorganisms, leading to an increase in butyrate production in the rumen, assuming that the Mg difference with significant correlation with Mg in first evaluation may also be involved in these situations. 

In the present study, significant correlation was observed between the AMH ratio and both the TP and A/G ratio derived from all 43 cows during the first evaluation. Additionally, significant differences were also observed in TP concentration between pre- and postpartum periods of both H- and L-AMH groups, as well as Alb concentration and A/G ratio during the postpartum period between the H- and L-AMH groups. This demonstrates a significant increase in globulin concentration in the L-AMH group. Alb is not only a metabolic indicator of protein status of cattle, but also a negative APP whose concentration falls gradually during the occurrence of infectious and inflammatory diseases [42]. Although multiple traits should be considered to assess protein status, such as BUN, creatinine, TP, and Alb [36], we identified no significant differences in BUN, AST, GGT, Bil, IGF1, NEFA, T-Cho, Glu, and TG between the H- and L-AMH groups in the present study. Characteristics of APPs such as increased HG and decreased Alb and PON concentrations have been reported in cows with postpartum uterine and systemic infections [3,43,44]. In cattle, the half-life of Alb is 16.5 days. Thus, hypoalbuminemia due to inflammation will take several weeks to develop [45]. Although postpartum (D28) clinical traits of inflammation, such as SCC, PMNL, and VDS, were not different between H- and L-AMH groups in both the uterus and mammary glands in the present study, we assumed that Alb concentration might be a sensitive parameter for the inflammation status. Considering the half-life of Alb, further experiments including more time points are needed to support this hypothesis.

SAA is an APP that is mainly synthesized in the liver; its synthesis is induced via inflammatory cytokines such as IL-1, IL-6, and TNF-α [22]. SAA and HG are the most prominent APPs in cows, produced in the liver and other tissues, including uterus [22,23]. Zhang et al. [23] recently indicated that endometrial local expression of SAA is more significant than HG as a potential biomarker to assess the severity of endometritis in dairy cows. Therefore, in the present study, we included the SAA concentration instead of HG as a parameter for inflammation status of the genital tract. Additionally, it has been reported that endometritis at 42 days postpartum was associated with lower concentrations of Alb and a lower A/G ratio throughout the transition period [41]. Our results from the first evaluation clearly indicated a significant negative correlation between SAA and the AMH ratio at D10 and correlated at D28 from all 43 cows that were examined. Additionally, we observed significant differences in both SAA and A/G ratio (possibly due to high γ-globulin concentration) in the postpartum period between the H- and L-AMH groups. This suggests that SAA and the A/G ratio might be more sensitive measures to monitor the inflammation status of postpartum dairy cows. Previously, it has been reported that factors such as nutrition (NEFA), hormones (including endocrine disrupting chemicals), and diseases (e.g., granulosa-theca cell tumor and mastitis) influence the ovarian reserve of small antral follicles, and hence, the AMH concentration of female animals [14,35,37,46]. To identify possible regulatory factors of AMH, we evaluated the change in AMH levels during the perinatal period and showed that the inflammation status, possibly reflected by an increase in SAA and a decrease in the A/G ratio, was associated with a decrease in serum AMH, and thus might have affected the dynamics of AMH-producing follicles. Interestingly, during the first evaluation, a positive correlation was observed between Mg concentration and the AMH ratio at D10 and D28, which was concomitant with the relationship between inflammation parameters such as the AG ratio and SAA. Previous reports on human cases have indicated that subclinical Mg deficiency caused by low dietary intake often occurring in the population is a predisposing factor for chronic inflammatory stress, which is conductive for chronic disease [47]. Therefore, in cattle, as in humans, Mg levels may be an important factor in monitoring postpartum chronic inflammatory conditions, and further studies are needed to clarify the significance.

Previous studies have shown that blood AMH concentration in cattle is characteristic of individual cows, and is a reliable endocrine marker for the size of the dynamic reserve of small antral follicles [48], and of the number of ovulation events and embryos produced in response to superovulation or OPU-IVP [13,14,15,34,36]. Additionally, studies with metadata from dairy farms have demonstrated the relationship between AMH concentration and reproductive fertility of dairy herds, suggesting that AMH is a reliable biomarker for the genetic improvement of reproductive potential [1,2]. As shown in Figure 3, in the present study, cows in the H-AMH (*n* = 11), HM-AMH (*n* = 11), and LM-AMH (*n* = 10) groups had increased AMH serum concentration from D-14 to D28, but the ones in the L-AMH (*n* = 11) group had decreased (7/11, 63%) or unchanged (4/11, 36%) AMH serum concentration. Therefore, AMH concentrations were increased in a large majority of postpartum cows compared with those in the prepartum period, suggesting that the dynamics of the AMH-secreting follicles was activated after calving in these cows. In the L-AMH group, AMH levels did not increase, indicating that follicular dynamics were affected, but the mechanisms underlying these changes in AMH concentration warrant further investigations. As shown in Table 6, no significant differences were observed between the H-AMH and L-AMH groups in the traits related to the postpartum resumption of ovarian and/or follicular function/activity. Both the low number of pregnant cows within 200 days postpartum and the high culling rate observed in the L-AMH group may be associated with its systemic inflammation status, which has not allowed us to conclude anything about the potential consequences of these changes on the reproductive efficiency of the cows, due to the low number of cows that were finally inseminated in this group.

The limitations of this study are as follows: First, the sample size was small; the post hoc power analysis showed that the power for detecting differences with large effect size was 0.607. Therefore, although the items that were found to be significant with a larger effect size difference can still be guaranteed to be statistically significant, the non-significant items do not necessarily negate the difference between groups, and the insufficient sample size should be taken into consideration. Second, we were not able to conduct sufficient multivariate analysis to include all confounding factors as covariates in the analysis. In this study, we were able to take into account some of the effects of farm and lactation number in the mixed model analysis, but we were not able to correct for other possible confounding factors such as feed intake, energy balance and the wide parity range. Finally, some of the test items had CV values exceeding 0.2, and it is necessary to secure a more accurate measurement system for future studies.

## 5. Conclusions

The results of our field study illustrated a relationship between changes in AMH concentration, especially reflected in the AMH ratio during the perinatal period, and the systemic inflammation status of dairy cows. We assume that certain inflammation status during the early postpartum period may decrease AMH levels. Whether it subsequently affects the reproductive prognosis of postpartum cows needs further investigation. Elucidating the mechanism of perinatal AMH changes may help improve reproductive efficacy in the dairy industry.

## Figures and Tables

**Figure 1 animals-11-01241-f001:**
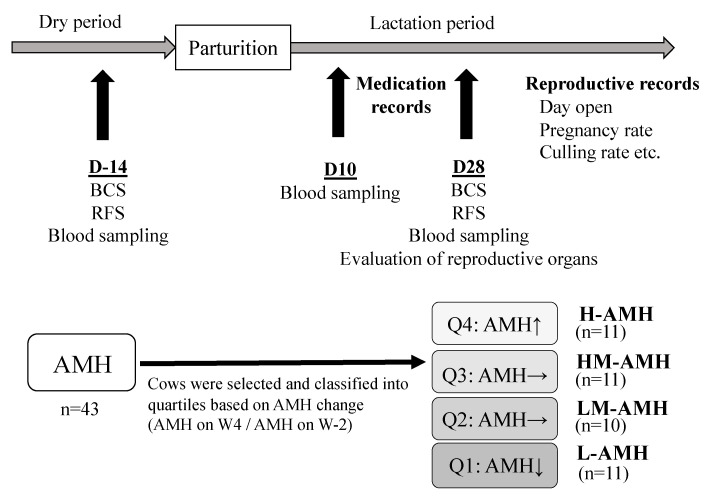
Experimental protocol for samplings and classification/selection of postpartum cows based on the perinatal change of anti-Müllerian hormone (AMH) concentration. BCS: body condition scoring; RFS: rumen fill score; D-14: 14 days before parturition; D10: 10 days after parturition; D28: 28 days after parturition.

**Figure 2 animals-11-01241-f002:**
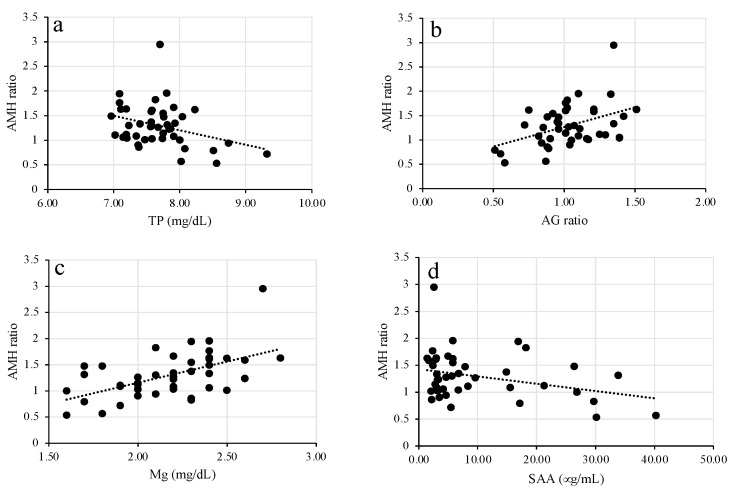
Representative significant correlation between the AMH ratio and (**a**) TP at D28, (**b**) A/G ratio at D28, (**c**) Mg at D10, and (**d**) SAA at D10.

**Figure 3 animals-11-01241-f003:**
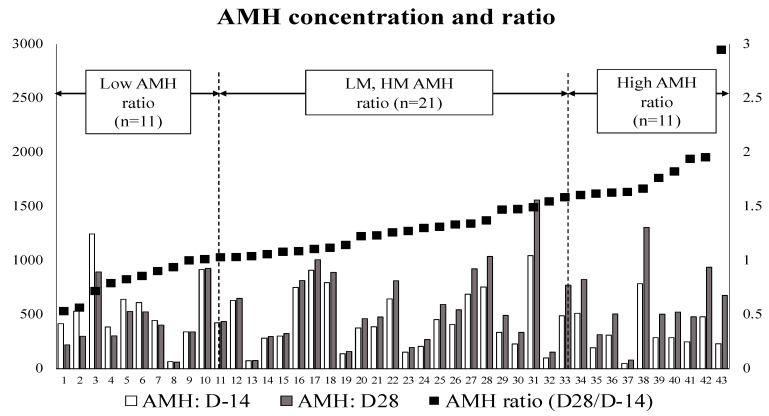
AMH concentration of prepartum (14 days before parturition: D-14) and postpartum (28 days after parturition: D28) cows, and the AMH ratio (D28/D-14) of each examined cow (*n* = 43). Low-AMH ratio: lowest 25% of cows (*n* = 11). LM (low-middle)- and HM (high-middle)-AMH ratios: both comprising 50% of cows (*n* = 21). High-AMH ratio: higher 25% of cows (*n* = 11). AMH concentrations are depicted as bars and expressed in pg/mL, relative the left ordinate scale. Values of AMH ratio are depicted as points, relative to the right ordinate scale.

**Figure 4 animals-11-01241-f004:**
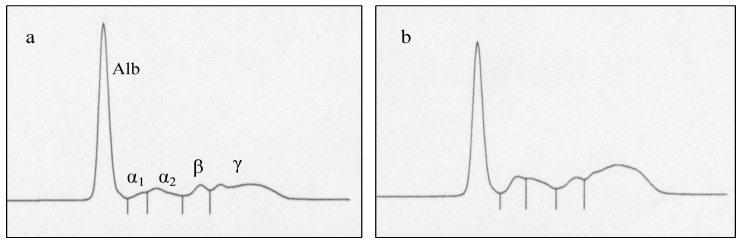
Results of protein electrophoresis of representative sera derived from (**a**) one cow from the high-AMH group (A/G ratio: 2.9) and (**b**) one cow from the low-AMH group (A/G ratio: 0.5) at D28. Alb: Albumin; α_1_: α_1_-globulin; α_2_: α_2_-globulin; β: β-globulin; γ: γ-globulin.

**Table 1 animals-11-01241-t001:** Pearson’s correlation coefficients (r) for relationships between the AMH ratio (AMH at D28/AMH at D-14) and biochemical and inflammation parameters with significant correlations in all 43 cows examined.

Parameter	D-14		D10		D28	
	*r*	*p*	*r*	*p*	*r*	*p*
TP	−0.4102	0.0063	−0.0597	0.7039	−0.3305	0.0304
A/G ratio	0.231	0.1362	0.3877	0.0102	0.435	0.0036
Mg	0.0872	0.5783	0.5512	<0.001	0.3958	0.0086
SAA	−0.0344	0.8265	−0.3161	0.039	−0.2880	0.0611
NEFA	0.1679	0.2819	0.414	0.0058	0.4034	0.0073

**Table 2 animals-11-01241-t002:** Anti-Müllerian hormone (AMH) concentration (pg/mL) in each classified cow group during pre- and postpartum samplings, and the AMH ratio of each group.

	Prepartum (D-14)	Postpartum (D28)	AMH Ratio
	AMH	AMH	(D28/D-14)
Q1: L-AMH (*n* = 11)	547.9 ± 94.5	449.6 ± 79.7	0.83 ± 0.05
Q2: LM-AMH (*n* = 10)	465.1 ± 91.3	516.4 ± 99.9	1.11 ± 0.02
Q3: HM-AMH (*n* = 11)	456.7 ± 89.4	629.4 ± 128.3	1.38 ± 0.03
Q4: H-AMH (*n* = 11)	351.9 ± 60.4	629.7 ± 99.1	1.83 ± 0.12

Cows (*n* = 43) were classified into quartiles (Q1, Q2, Q3, and Q4) based on their change in AMH pre- and postpartum concentrations (D28/D-14). L: low; LM: low-medium; HM: high-medium; H: high; D-14: 2 weeks before parturition; D28: 4 weeks after parturition.

**Table 3 animals-11-01241-t003:** Results of body condition score (BCS), rumen fill score (RFS), and serum biochemical analyses (mean ± SEM) of H- and L-AMH cows at pre- (D-14) and postpartum (D28) periods.

	Prepartum (D-14)						Postpartum (D28)					
	H-AMH			L-AMH			H-AMH			L-AMH		
BCS	3.2	±	0.1 ^c^	3.4	±	0.1 ^d^	2.8	±	0.1	2.9	±	0.1
RFS	3.5	±	0.2	3.9	±	0.2	3.1	±	0.2	3.0	±	0.3
AST (U/L)	59.2	±	3.5	52.6	±	3.5	73.7	±	5.1	64.7	±	4.8
GGT (U/L)	17.2	±	1.9	15.7	±	1.1	22.3	±	1.1	29.9	±	7.9
BUN (mg/dL)	8.3	±	0.6	8.2	±	0.4	10.2	±	0.8	10.1	±	1.1
Bil (mg/dL)	0.18	±	0.02	0.13	±	0.01	0.19	±	0.01	0.15	±	0.02
NEFA (mmol/L)	169.2	±	29.4	120.6	±	11.0	325.0	±	41.6	241.6	±	31.9
T-Cho (mg/dL)	69.7	±	4.0	76.2	±	4.3	172.7	±	10.5	155.7	±	8.4
Glu (mg/dL)	56.4	±	4.1	62.9	±	3.2	48.2	±	3.3	57.6	±	4.7
TG (mg/dL)	15.7	±	0.8	17.2	±	1.4	5.5	±	0.3	5.9	±	0.3
Ca (mg/dL)	9.3	±	0.1	9.4	±	0.1	9.5	±	0.2	9.3	±	0.1
iP (mg/dL)	5.9	±	0.3	5.8	±	0.2	5.7	±	0.2	5.4	±	0.3
Mg (mg/dL)	2.4	±	0.1	2.3	±	0.04	2.6	±	0.1 ^a^	2.3	±	0.1 ^b^
Fe (μg/dL)	170.1	±	9.5	151.6	±	9.2	125.5	±	6.7	101.6	±	12.4
3-HB (μmol/dL)	643.7	±	25.3	645.5	±	47.5	1504.4	±	229.5 ^a^	844.5	±	184.2 ^b^
TP (mg/dL)	7.0	±	0.2 ^a^	7.6	±	0.2 ^b^	7.5	±	0.1 ^a^	8.1	±	0.2 ^b^
Alb (mg/dL)	3.7	±	0.1	3.7	±	0.1	4.0	±	0.1 ^a^	3.6	±	0.1 ^b^
A/G ratio	1.15	±	0.07	1.01	±	0.09	1.14	±	0.06 ^a^	0.84	±	0.06 ^b^
IGF1 (ng/mL)	31.8	±	6.4	37.9	±	5.9	42.3	±	5.4	39.3	±	4.9

AST: aspartate aminotransferase; GGT: γ-glutamyltransferase; BUN: urea nitrogen; Bil: bilirubin; NEFA: non-esterified fatty acid; T-Cho: total cholesterol; Glu: glucose; TG: triglyceride; Ca: calcium; iP: inorganic phosphorus; Mg: magnesium; Fe: iron; 3-HB: 3-hydroxybutyric acid; TP: total protein; Alb: albumin; A/G ratio: albumin/globulin ratio; IGF1: insulin like growth factor-1. ^a;b^
*p* < 0.05, ^c;d^
*p* < 0.1.

**Table 4 animals-11-01241-t004:** Results of serum protein fractions and serum amyloid A (SAA) concentration of H- and L-AMH cows in pre- (D-14) and postpartum (D28) periods (mean and 95%CI).

	Prepartum (D-14)				Postpartum (D28)			
	H-AMH Group		L-AMH Group		H-AMH Group		L-AMH Group	
Total protein (mg/dL) ^(1)^	7.1	6.4–7.9 *	7.6	6.9–8.4 *	7.6	7.2–8.0 ^#^	8.1	7.7–8.5 ^#^
Albumin	3.6	3.0–4.2	3.6	3.1–4.2	3.9	3.4–4.3 ^##^	3.5	3.0–4.0 ^##^
α_1_-globulin	0.23	0.20–0.26 **	0.26	0.24–0.29 **	0.29	0.21–0.37	0.33	0.25–0.41
α_2_-globulin	0.60	0.52–0.69	0.62	0.53–0.71	0.64	0.59–0.69	0.66	0.61–0.71
β-globulin	0.67	0.54–0.81	0.83	0.69–0.96	0.62	0.48–0.76 ^###^	0.81	0.67–0.94 ^###^
γ-globulin	2.2	1.6–2.8	2.2	1.6–2.9	2.0	1.5–2.6 ^####^	2.6	2.1–3.2 ^####^
SAA (μg/mL)	10.9	0.0–35.0	10.9	0.0–35.0	3.5	0.0–17.3 ^#####^	22.4	9.5–35.3 ^#####^

^(1)^ A linear mixed model with farm as the random factor and lactation number as the covariate. Data are expressed as mean and 95% confidence interval (CI). * *p* = 0.031, ** *p* = 0.032, ^#^
*p* = 0.023, ^##^
*p* = 0.007, ^###^
*p* = 0.027, ^####^
*p* = 0.044, ^#####^
*p* = 0.022.

**Table 5 animals-11-01241-t005:** Results of somatic cell count (SCC) in milk and postpartum reproductive traits for recovery of uterus and ovarian function at D28.

	H-AMH		L-AMH	
SCC (/μL)	143.1	[0.0–1069.9]	728.4	[0.0–1632.1]
	(DIM: 35.3 [29.4–41.3]) *		(DIM: 30.5 [24.6–36.5]) *	
PMNL (%)	15.9	[2.8–29.0]	17.8	[4.7–30.9]
VDS	0.63	[0.04–1.22]	0.56	[0.05–1.06]
CL rate (%)	27.0 [0.0–59.0] (3/11)		45.8 [13.7–77.8] (5/11)	

SCC: mean from first and second sets of postpartum SCC data; DIM: days in milk; PMNL: polymorphonuclear leukocyte; VDS: vaginal discharge score; CL rate: % of cows that confirmed the presence of a corpus luteum in the ovary. PMNL, VDS, and CL confirmations were conducted at D28 blood sampling. A linear mixed model or generalized linear mixed model (link function: logit) with the farm as the random factor and lactation number as the covariate. Data are expressed as the mean and 95%CI. * The mean DIM between the H- and L-AMH groups was not significantly different (*p* = 0.250).

**Table 6 animals-11-01241-t006:** Results of the postpartum reproductive records of H-AMH and L-AMH cows.

	No. of Medication Treatment	Calving to FAI Interval (Days)	Pregnancy Rate of FAI	Days Open	No. of AI	No. of Pregnant Cows within 200 DIM	Culling Rate
H-AMH (*n* = 11)	0.8 [0.0–3.2] * (*n* = 3)	105.7 [84.4–127.0]	5/11 50.0% [11.8–88.3]	147.1 [114.3–179.9]	1.9 [0.3–3.4]	9/11 ** 78.2% [42.8–100.0]	1/11 *** 20.7% [0.0–56.8]
L-AMH (*n* = 11)	3.9 [1.5–6.3] * (*n* = 9)	89.8 [64.8–114.7]	3/8 ^#^ 59.9% [5.7–100.0]	133.3 [86.8–179.7]	1.1 [0.0–2.8]	4/11 ** 35.8% [1.8–69.9]	6/11 *** 58.9% [23.0–94.7]

FAI: first artificial insemination; DIM: days in milk. A linear mixed model or generalized linear mixed model (link function: logit) with the farm as the random factor and lactation number as the covariate. Data are expressed as the mean and 95%CI. * *p* = 0.072, ** *p* = 0.039, *** *p* = 0.010. ^#^ Three cows were culled before conducting the first artificial inseminations. Medication treatment for the following conditions: retained placenta, dystocia, milk fever, ketosis, uterine torsion, udder edema, mastitis, indigestion, and arthritis.

## Data Availability

Not applicable.

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
