# Peer review of "Association between Anti-Müllerian Hormone Concentration and Inflammation Markers in Serum during the Peripartum Period in Dairy Cows"

_animals, 2021, doi:10.3390/ani11051241_

Round 1

Reviewer 1 Report

None

Author Response

Thank you very much for your kind reviewing.

Reviewer 2 Report

Authors have addressed my concerns and improved the manuscript.

Author Response

(The authors gave the same response as above.)

Reviewer 3 Report

I have no more remarks. In my opinion the manuscript is suitable to be published in Animals

Author Response

Thank you very much for your kind reviewing.

This manuscript is a resubmission of an earlier submission. The following is a list of the peer review reports and author responses from that submission.

Round 1

Reviewer 1 Report

Review of the manuscript animals-1100797

The manuscript entitled ‘Association between anti-Müllerian hormone concentration and inflammation markers in serum during the peripartum period in dairy cows’ presents very interesting and new data concerning metabolic and inflammatory status in relation to changes in anti-Müllerian hormone levels before and after parturition. The objectives of the study and study design are rational and the methods are properly chosen. The results are clearly presented with the use of tables and figures.

However, in my opinion there are three major remarks that should be clarified.

  • Line 123 – Did the authors include all cows that delivered during study period or cows were selected? If so what were the selection criteria and could they influence on the results?
  • Line 138 – ‘The parity of the cows ranged from one to seven’. What was the distribution of the cows according to the lactation number. It is known that primiparous and multiparous cows differ from metabolic and reproductive point of view, thus lactation number is a very crucial trait speaking about cows’ health.
  • Lines 437-441 – ‘Although postpartum (D28) clinical traits of inflammation such as SCC, PMNL, and VDS were not different between the H- and L-AMH groups in both the uterus and mammary gland in the present study, we concluded that Alb concentration may be a sensitive parameter for the inflammation status in the present study.’ In my opinion this conclusion is not justified by the data so should be deleted or re-written. The authors could conclude their results in that way having more time points and samples to measure ALB. Having in mind their half-life the sample should be taken for a longer period.

The minor remarks are listed below:

  • Line 113 – replace ‘determine’ with ‘describe’
  • Line 152 – protocol not protocols
  • Line 156 – to clarify I would explain also shortcut ‘D10”
  • Line 267 – ‘No significant correlation was observed between any of the examined parameters at D-14.’ , but it is shown in the Table 1 that there is a significant difference in TP at that time – please explain
  • Table 2 – shows data concerning concentration but there are no units (they are in the Figure 3)
  • Line 318 – concentrations
  • In my opinion Figure 4 is not necessary because the Table 4 presents these data. If the authors decide that should be included there must be description which time point this data refer to.
  • Line 370 – retained placenta
  • Lines 407-408 – ‘Thus, we determined that the cows in the H-407 AMH group might not be under the NEB condition compared to the cows in the L-AMH 408 ‘ – analizing data it seems that none of the group is under the NEB – please explain

Taking into account the above the manuscript is not suitable to be published in that form and should be corrected.

Author Response

Ms. Ref. No.: animals-1100797

Title: Association between anti-Müllerian hormone concentration and inflammation markers in serum during the peripartum period in dairy cows

Journal: Animals

We thank all the Reviewers for their constructive comments. We have revised the manuscript in accordance with the suggestions of Reviewers as follows: Both the revised sections of the manuscript and the responses to the Reviewers below are marked in red for Reviewer 1, blue for Reviewer 2 and green for Reviewer 3.

  • Reviewer 1:
    Line 123 – Did the authors include all cows that delivered during study period or cows were selected? If so what were the selection criteria and could they influence on the results?

All cows used in the present study were selected randomly after clinically normal calving from four dairy herds during the experimental periods. Accordingly, we have added this information in the manuscript. Lines 121-122.

Line 138 – ‘The parity of the cows ranged from one to seven’. What was the distribution of the cows according to the lactation number. It is known that primiparous and multiparous cows differ from metabolic and reproductive point of view, thus lactation number is a very crucial trait speaking about cows’ health.

According to your suggestion, we have added the results of the mean lactation number ± SEM of the examined four groups with no significant differences among them. Lines 136-138.

Lines 437-441 – ‘Although postpartum (D28) clinical traits of inflammation such as SCC, PMNL, and VDS were not different between the H- and L-AMH groups in both the uterus and mammary gland in the present study, we concluded that Alb concentration may be a sensitive parameter for the inflammation status in the present study.’ In my opinion this conclusion is not justified by the data so should be deleted or re-written. The authors could conclude their results in that way having more time points and samples to measure ALB. Having in mind their half-life the sample should be taken for a longer period.

Thank you very much for your insightful comment. According to the comments, we have revised the manuscript, adding not a conclusive sentence but an assumption, as follows: “…, we assumed that Alb concentration might be a sensitive parameter for the inflammation status. Considering the half-life of Alb, further experiments including more time points are needed to support this hypothesis.” Lines 445-448.

  • Line 113 – replace ‘determine’ with ‘describe’

Accordingly, this word was replaced. Line 112.

  • Line 152 – protocol not protocols

Accordingly, we corrected the word. Line 151.

  • Line 156 – to clarify I would explain also shortcut ‘D10”

We have added the explanation of D10 to the legend of Figure 1.

  • Line 267 – ‘No significant correlation was observed between any of the examined parameters at D-14.’ , but it is shown in the Table 1 that there is a significant difference in TP at that time – please explain

Thank you very much for this important remark and we apologize for overlooking this. We have revised the sentence as follows: “except for TP”. We could speculate that a high TP content before parturition might be predictive of a decrease in AMH concentrations after calving. Further experiments with a higher number of cows are required to support this assumption and identify the concerned proteins. Line 269.

  • Table 2 – shows data concerning concentration but there are no units (they are in the Figure 3)

According to your comment, we have added the units to the title of Table 2.

  • Line 318 – concentrations

We have revised accordingly. Line 322.

  • In my opinion Figure 4 is not necessary because the Table 4 presents these data. If the authors decide that should be included there must be description which time point this data refer to.

Table 4 presents the data (mean ± SEM) of protein fractions obtained in the two groups of animals at D-14 and D28. Figure 4 illustrates the protein electrophoresis results of two representative sera of each group at D28. We kindly propose to keep this Figure and, according to your suggestion, we have added more information to the legend of Figure 4.

  • Line 370 – retained placenta

Thank you for this remark. We have revised accordingly. Line 380.

  • Lines 407-408 – ‘Thus, we determined that the cows in the H-407 AMH group might not be under the NEB condition compared to the cows in the L-AMH 408 ‘ – analizing data it seems that none of the group is under the NEB – please explain

We agree with this comment and have changed the sentence to clarify the text. Lines 413-416.

Reviewer 2 Report

The design of this study is problematic, 43 cows from 4 different farms that had different feeding management. Also, there is no direct data on feed intake or energy balance that can affect reproductive system. The large variation in lactation number, 1-9, clearly affected the AMH values but this is not included in the statistical model and no information is provided in the text on the distribution of the age of cows in Q1-Q4. Also, the effect of farm is not shown in data or in statistical analysis. All data on reproductive success, medication and culling is irrelevant for such a limited number of cows, and does not have scientific merit. Overall, this manuscript lacks vital information on the energy balance of the cows, the effect of lactation number and farm. the limited number of cows from 4 farms are a main concern, and the statistical analysis as shown is not appropriate.

Author Response

Ms. Ref. No.: animals-1100797

Title: Association between anti-Müllerian hormone concentration and inflammation markers in serum during the peripartum period in dairy cows

Journal: Animals

We thank all the Reviewers for their constructive comments. We have revised the manuscript in accordance with the suggestions of Reviewers as follows: Both the revised sections of the manuscript and the responses to the Reviewers below are marked in red for Reviewer 1, blue for Reviewer 2 and green for Reviewer 3.

Reviewer 2:
The design of this study is problematic, 43 cows from 4 different farms that had different feeding management. Also, there is no direct data on feed intake or energy balance that can affect reproductive system. The large variation in lactation number, 1-9, clearly affected the AMH values but this is not included in the statistical model and no information is provided in the text on the distribution of the age of cows in Q1-Q4. Also, the effect of farm is not shown in data or in statistical analysis. All data on reproductive success, medication and culling is irrelevant for such a limited number of cows, and does not have scientific merit. Overall, this manuscript lacks vital information on the energy balance of the cows, the effect of lactation number and farm. the limited number of cows from 4 farms are a main concern, and the statistical analysis as shown is not appropriate.

Thank you very much for your important remarks. We have modified a mixed model, in which the farm was the random factor and the lactation number was the covariate in the comparison between groups (H-AMH vs. L-AMH) for outcome parameters. Since it is difficult to add feed intake and energy balance as covariates in the number of cases in this study, only the lactation number, which is considered to be the strongest confounding factor, was added as a covariate. The above changes are described in Materials and Methods and Results. Furthermore, we have added the research limitation that the sample size was not enough to add all confounding factors to the covariates.

Reviewer 3 Report

The aim of the study was to relate perinatal CHANGES in serum AMH concentrations (reflecting changes in the dynamics of preantral and early antral ovarian follicle population) with metabolic, immunological, and health status markers as well as reproductive performance.

This is undoubtedly an original/novel scientific approach but it is not clear what supports the experimental hypothesis of the study; i.e. metabolic, immunological, and health status conditions might influence high AMH-producing ovarian follicle population (and serum AMH concentrations) and through it also influence reproductive performance.

The experimental design however is adequate, as a first approach, to relate perinatal CHANGES in serum AMH (reflecting dynamics of AMH-producing follicles) with physiological/pathological conditions in dairy cows.

General comments.

INTRODUCTION

The relationship between AMH levels and ovarian follicle populations was referred to by the authors. Possible changes in serum AMH from gestation to the postpartum period, mostly reflecting numerical changes in the population of high AMH-producing follicles, were also mentioned.

Direct or indirect evidence of metabolic, immunological, or health status conditions, detrimental for reproductive performance/physiology, influencing the absolute number or dynamics of AMH-producing follicles is not presented/commented.

Justification/support of the experimental hypothesis  MUST BE IMPROVED.

DISCUSSION/CONCLUSIONS

Interpretation of the results must consider:

-The response variable examined was perinatal "change" in AMH (dynamics of AMH-producing follicles) and not actual/absolute AMH serum concentrations (total number of AMH-producing follicles); this must be clearly stated throughout discussion (i.e. avoid phrases like "factor affecting AMH concentrations and postpartum ovarian reserves" L-464, which coul be missinterpreted). 

-H-AMH group included cows increasing their AMH serum concentration (and number of AMH-producing follicles) from pp days -10 to +28 and L-AMH group included cows decreasing (7/11, 63%) or almost maintaining (4/11, 36 %) those concentrations.

-Correlation analysis only measures the amount of association between variables and not a cause-effect relationship.

-The observed correlation coefficients were of medium to low value (<0.5); thus the association between variables is not strong.

-Results on Albumin, Albumin/Globulin ratio, and SAA do not conclusively indicate a different "inflammation status" related to the perinatal change trend of serum AMH concentrations as is ambiguously stated in the conclusions; "excessive inflammation during the early postpartum period, L491-492"

-No effect on reproductive efficiency was apparently present; pregnancy rate within 200 DMI was not correctly estimated (in the L-AMH group denominator for PR had to be 8, not 11 and PR 50 %, not 36.4 %, thereafter, nonsignificant differences exist between H- and L-AMH groups)

-Culling rate might be general culling rate (voluntary and involuntary) and not neccesarily associated with reproductive failure; indicate if the culling rate is only related to reproductive failure.

Discussion and conclusions MUST BE IMPROVED; focused on results and without ambiguities. 

Author Response

Ms. Ref. No.: animals-1100797

Title: Association between anti-Müllerian hormone concentration and inflammation markers in serum during the peripartum period in dairy cows

Journal: Animals

We thank all the Reviewers for their constructive comments. We have revised the manuscript in accordance with the suggestions of Reviewers as follows: Both the revised sections of the manuscript and the responses to the Reviewers below are marked in red for Reviewer 1, blue for Reviewer 2 and green for Reviewer 3.

Reviewer 3:

INTRODUCTION

The relationship between AMH levels and ovarian follicle populations was referred to by the authors. Possible changes in serum AMH from gestation to the postpartum period, mostly reflecting numerical changes in the population of high AMH-producing follicles, were also mentioned.

Direct or indirect evidence of metabolic, immunological, or health status conditions, detrimental for reproductive performance/physiology, influencing the absolute number or dynamics of AMH-producing follicles is not presented/commented.

Justification/support of the experimental hypothesis MUST BE IMPROVED.

According to the suggestions/comments, we have added a sentence before mentioning the experimental hypothesis as follows; “The growth of small antral follicles, which produce AMH, can be influenced by changes in metabolic hormone concentrations induced by acute changes in nutrient intake (Webb et al., 2004; Webb et al, 2007; Scaramuzzi et al., 2010). These follicles are also capable of responding to inflammatory mediators, which may perturb their development (Sheldon et al., 2014; Gilbert, 2019). However, direct evidence of immunological, or other health conditions, detrimental to reproductive performance and influencing the number or dynamics of AMH-producing follicles, has not been demonstrated so far. Moreover, to our knowledge, no reports are available on the relationship between the changes in serum AMH concentration and inflammation status derived from both systemic NEB and bacterial infections of the genital tract that occur during perinatal period in cows.” Lines 93-102.

DISCUSSION/CONCLUSIONS

Interpretation of the results must consider:

-The response variable examined was perinatal "change" in AMH (dynamics of AMH-producing follicles) and not actual/absolute AMH serum concentrations (total number of AMH-producing follicles); this must be clearly stated throughout discussion (i.e. avoid phrases like "factor affecting AMH concentrations and postpartum ovarian reserves" L-464, which could be misinterpreted). 

We agree with your comment. We have changed the sentence in order to make the text clearer. The revised sentence is as follows: “…that the inflammation status, possibly reflected by an increase in SAA and a decrease in the A/G ratio, was associated with a decrease in serum AMH and thus might have affected the dynamics of AMH-producing follicles.” Lines 468-471.

-H-AMH group included cows increasing their AMH serum concentration (and number of AMH-producing follicles) from pp days -10 to +28 and L-AMH group included cows decreasing (7/11, 63%) or almost maintaining (4/11, 36 %) those concentrations.

Thank you very much for this important remark. We have included this in the Discussion section as follows; “As shown in Figure 3, in the present study, cows in H-AMH (n=11), HM-AMH (n=11) and LM-AMH (n=10) groups included had increased AMH serum concentration from D-14 to D28, but the ones in the L-AMH (n=11) group included had decreased (7/11, 63%) or unchanged (4/11, 36 %) AMH serum concentrations. Therefore, AMH concentrations were increased in a large majority of postpartum cows compared with those in the prepartum period, suggesting that the dynamics of the AMH-secreting follicles was activated after calving in these cows. In the L-AMH group, AMH levels did not increase, indicating that follicular dynamics were affected, but the mechanisms underlying these changes in AMH concentration warrant further investigations.” Lines 485-494.

-Correlation analysis only measures the amount of association between variables and not a cause-effect relationship.

-The observed correlation coefficients were of medium to low value (<0.5); thus the association between variables is not strong.

We agree with this comment. In the present study, we have tried to discuss the obtained results regarding the observed significant correlations.

-Results on Albumin, Albumin/Globulin ratio, and SAA do not conclusively indicate a different "inflammation status" related to the perinatal change trend of serum AMH concentrations as is ambiguously stated in the conclusions; "excessive inflammation during the early postpartum period, L491-492"

Accordingly, we have replaced “excessive” with “certain inflammation status” Line 506.  

-No effect on reproductive efficiency was apparently present; pregnancy rate within 200 DMI was not correctly estimated (in the L-AMH group denominator for PR had to be 8, not 11 and PR 50 %, not 36.4 %, thereafter, nonsignificant differences exist between H- and L-AMH groups)

We apologize for the mistake in estimating the pregnancy rate in the L-AMH group. We agree that there was no difference in the reproductive efficiency between groups, and we have changed the relevant text in the abstract, results and discussion/conclusion sections. Thank you very much for this insightful remark. We have revised the column as “pregnant cows” Table 6, Lines 43-45, Lines 496-500.

-Culling rate might be general culling rate (voluntary and involuntary) and not necessarily associated with reproductive failure; indicate if the culling rate is only related to reproductive failure.

Although the reasons of culling were obscure in the present study, significant difference was observed between H-AMH and L-AMH groups, which may indirectly reflect the difference of systemic inflammation status of the two groups. Therefore, as we mentioned in the Discussion section, we would like to keep the culling rate results to propose future research issues.    

Discussion and conclusions MUST BE IMPROVED; focused on results and without ambiguities. 

Thank you very much for this important remark. As we responded above, we have revised the Discussion and Conclusion sections according to all the remarks/comments. Lines 506-510.
